# Differential DNA Methylation from Autistic Children Enriches Evidence for Genes Associated with ASD and New Candidate Genes

**DOI:** 10.3390/brainsci13101420

**Published:** 2023-10-07

**Authors:** Mirna Edith Morales-Marín, Xochitl Helga Castro Martínez, Federico Centeno Cruz, Francisco Barajas-Olmos, Omar Náfate López, Amalia Guadalupe Gómez Cotero, Lorena Orozco, Humberto Nicolini Sánchez

**Affiliations:** 1Laboratorio de Genómica de Enfermedades Psiquiátricas y Neurodegenerativas, Instituto Nacional de Medicina Genómica, Mexico City 14610, Mexico; xcastro@inmegen.gob.mx (X.H.C.M.); hnicolini@inmegen.gob.mx (H.N.S.); 2Laboratorio de Inmunogenómica y Enfermedades Metabólicas, Instituto Nacional de Medicina Genómica, Mexico City 14610, Mexico; fcenteno@inmegen.gob.mx (F.C.C.); fbarajas@inmegen.gob.mx (F.B.-O.); lorozco@inmegen.gob.mx (L.O.); 3Hospital de Especialidades Pediátricas, Tuxtla Gutiérrez 29045, Mexico; axlnafate@gmail.com; 4Centro de Investigación en Ciencias de la Salud, Unidad Santo Tomás, Instituto Politécnico Nacional, Mexico City 07738, Mexico; amaliagpegomez@prodigy.net.mx; 5Grupo Médico Carracci, Mexico City 03740, Mexico

**Keywords:** ASD, gene methylation, Mexican children

## Abstract

The etiology of Autism Spectrum Disorders (ASD) is a result of the interaction between genes and the environment. The study of epigenetic factors that affect gene expression, such as DNA methylation, has become an important area of research in ASD. In recent years, there has been an increasing body of evidence pointing to epigenetic mechanisms that influence brain development, as in the case of ASD, when gene methylation dysregulation is present. Our analysis revealed 853 differentially methylated CpG in ASD patients, affecting 509 genes across the genome. Enrichment analysis showed five related diseases, including autistic disorder and mental disorders, which are particularly significant. In this work, we identified 64 genes that were previously reported in the SFARI gene database, classified according to their impact index. Additionally, we identified new genes that have not been previously reported as candidates with differences in the methylation patterns of Mexican children with ASD.

## 1. Introduction

Autism spectrum disorder (ASD) is defined as a neurodevelopmental disorder with characteristics such as difficulty with social interaction, poor communication skills and repetitive behaviors. The frequency is higher among males than it is among females (4:1), and its heritability is as high as 90%, depending on the population analyzed [1]. The high concordance between monozygotic twins shows a strong genetic influence. In Mexico, epidemiological studies are scarce, and it has been estimated to have a prevalence of 0.87%, which is similar to the world statistic [2]. ASD patients also present with a higher prevalence of other clinical manifestations, such as depression, ADHD, anxiety, gastrointestinal problems and sleep disorders, among others [3,4].

Cumulative evidence supports that genetic factors underlie ASD, although the molecular and cellular mechanisms are still unidentified. Single-nucleotide variants (SNV) are associated with ASD in genes that play a role in neurodevelopment, synapsis, ion transport and neurotransmission, i.e., *SYN1*, *SCN2A*, *CACNA1E*, *KCNQ3*, *SHANK3* and *GABRG3* [5]. Similarly, copy number variants (CNV) have shown an important effect on ASD susceptibility, and it is estimated to cause around 10% of cases [6,7]. Mutations in animal models of ASD-related genes showed a phenotype with disorders such as motor coordination, sensorial, weight loss, hydrocephalus, anxiety, repetitive behaviors and social deficits similar to those of ASD patients [8].

Despite the genetic evidence, there is minimal information on gene-altered functions in ASD and neurodevelopmental disorders. The evidence from exome sequencing has revealed de novo mutations and disruptive events in candidate genes [9,10]. However, the effects of most ASD-related gene variations are quite small.

Recent evidence shows that epigenetic regulation (DNA methylation, histone modifications and non-coding RNAs) may play an important role in ASD, integrating genetic and environmental factors affecting neurodevelopment [11].

Epigenetic mechanisms regulate gene expression without altering the DNA sequence, such as DNA methylation. In recent years, DNA methylation has been implicated in ASD pathophysiology, and it has been proposed that epigenetic alterations could influence neural programming during development and early life [12,13,14]. Studies in monozygotic discordant twins with ASD have identified CpG sites with altered DNA methylation in genes previously associated with ASD, describing epigenomic discordant patterns [15]. Subsequent studies have shown altered methylation in genes such as *OXTR*, *GAD1*, *EN2*, *RELN*, *MECP2*, *RUNX2* and *IMMPL2* [16,17]. CpGs with altered methylation in those genes have been proposed as molecular biomarkers for ASD. On the other hand, ASD animal models induced by valproic acid have shown exacerbated DNA demethylation, therefore, interfering with normal brain development correlating with the epidemiological data that showed an increased risk in children exposed during the first three months of their intrauterine life [18].

Although it has been pointed out that methylation patterns are tissue specific, the search for methylation markers has not been restricted to brain samples, which are only available in brain biobanks. Postmortem samples from children are particularly extremely difficult to obtain. On the other hand, molecular biomarker candidates should be from samples obtained using non-invasive methods. To solve these inconveniences, the alternatives are the use of peripheral tissues, such as blood and buccal cells [16,19,20,21]. The usefulness of buccal cells has been demonstrated in their application to establish methylation patterns and pediatric age markers [22,23]. Since the methylation patterns vary depending on the tissue, cell type and age, ASD risk biomarkers should either be consistently present and identifiable across different tissues or accurately reflect alterations in the target tissue [24].

In this work, we evaluated the DNA methylation profiles of a group of Mexican male children with an ASD diagnosis and controls in order to obtain CpG sites (CpGs) with altered methylation in pediatric patients using DNA samples from buccal swabs.

## 2. Materials and Methods

Study population: Twenty-nine cases were selected from a bigger sample established previously among CMIGA (Consorcio Mexicano de Investigación Genética en Autismo) patients. This heterogeneous sample was not selected for genetic features; the main characteristics are described in Table 1. The inclusion criteria for cases were as follows: being male, 3–7 years old and having a previous ASD diagnosis based on the DSM-5 (ADI–R instrument [25]), corroborated by trained specialists. For the seven controls with neurotypical development, we matched the cases by age (3–7 years old), sex (males) and nationality. A psychiatric scale was applied, the Social Responsiveness Scale (SRS), to exclude autistic symptomatology, and the criterium of having no psychiatric history was applied [26]. The exclusion criteria for the controls include being female, out of the age range diagnosed with neurodevelopmental disorders or psychiatric conditions. The controls were from a previous biobank established, the parents of whom were invited to participate prior to an informative talk in a public center or school. For methylation studies, it is important to match the samples by age because it is demonstrated that this factor influences gene methylation with age We excluded females to avoid the influence of the X chromosome. All the participants referred were the third generation of their family to be born in Mexico and corroborated their ancestry with the PCA plot. This study was approved by ethical committees in Grupo Médico Carracci in accordance with the Helsinki Declaration.

Sample collection and DNA extraction: Non-invasive samples were taken for this study using sterile buccal swabs to obtain epithelial cells via scraping. Immediately, nucleic acids were extracted with a Gentra Puregene kit (Qiagen, Hilden, Germany) following the manufacturer’s instructions. DNA concentration as well as 260/230 and 280/230 ratios were determined via spectrophotometry with a Nanodrop instrument (Thermofisher, Waltham, MA, USA). DNA integrity was visualized in 1% agarose gel.

Genome-wide DNA Methylation Profiling: DNA methylation profiles were generated with Illumina Infinium Human Methylation 450K BeadChip arrays (Illumina, San Diego, CA, USA). High-quality DNA was bisulfite converted using an EZ DNA methylation kit (Zymo Research, Irvine, CA, USA). Bisulfite-converted DNA was whole-genome amplified for 23 h, which was followed by end-point fragmentation. Fragmented DNA was precipitated, denatured and hybridized using BeadChips for 18 h at 48 °C. The BeadChips were washed, and the hybridized primers were extended and labeled prior to scanning the BeadChips using the Illumina iScan system.

We read methylation data from the raw IDAT files using the Chip Analysis Methylation Pipeline package (ChAMP v2.16.2) R package and calculated the β-value for each CpG as β = M/(M + U + α), where M and U represent methylated and unmethylated signal intensities at the specific site, respectively, and α is an arbitrary offset (usually 100) intended to stabilize the β-values where the fluorescent intensities are low. SNPs and probes with a low detection rate were removed from the analysis. To identify the differentially methylated regions (DMRs), we used the ChAMP package

For enrichment analysis, we used WebGestalt 2019 software, a functional enrichment analysis web tool. Its main function is to translate gene lists into biological insights. The parameters chosen were the organism of interest—Homo sapiens; the method of interest—ORA (over-representation analysis); and the functional database—disease. Parameters for the enrichment analysis: FDR method—Benjamini–Hochberg; significance level—FDR < 0.05.

## 3. Results

### 3.1. Differential DNA Methylation

We selected only male patients, as DNA methylation is affected by sex, not only in regard to sex chromosomes, but also autosomes. DNA methylation data were acquired with Illumina 450K microarrays, and differentially methylated CpGs (DMCs) were obtained by comparing the ASD patients’ profiles to the controls (*p* < 0.05; delta beta ≥ 0.10). A total of 853 DMCs were found between the ASD patients and controls, representing 84% hypomethylated (green dots) and 16% hypermethylated (red dots) ones, as shown in the volcano plot (Figure 1a and Appendix A). DMCs were located in 509 protein-coding genes, 278 intergenic regions and 15 non-coding RNA genes, which were distributed throughout the whole genome, as seen in the Manhattan plot (Figure 1a lower panel and Appendix A). Twenty-six genes showed more than one DMC. However, no region reached a level of significance to be considered as a DMR.

The most abundant DMCs were found in open sea regions (62%), followed by the shore (20%), shelf (10%) and island (8%) (Figure 1b upper panel). In Figure 1b, the bottom panel shows the gene position of the identified DMCs, which were the most abundant in the body of genes and intergenic regions, and the rest were distributed along other gene positions. The five most statistically significant DMCs were found within the genes *ISM1*, *PTPRG*, *SLITRK4*, *CAP2* and *CYP26C1*.

Most of the genes with DMCs only included one per gene, although 33 genes were found with more than one CpG.

In the heatmap, unsupervised hierarchical data clustering showed the separation of the subjects in each group, with the exception of one ASD patient who clustered within the control group when studying the top 50 most significant DMCs (Figure 1c).

### 3.2. Functional Analysis of DMCs

To identify the functional categories of genes with DMCs, gene enrichment analysis was performed. Five important categories were indicated using the software WebGestalt [27], and we grouped the statistically significant genes as follows: autistic disorder (AD) was the most identified disease and mental disorders term (MD), as well as aortic rupture, premature birth and musculoskeletal diseases (Figure 2). The genes within AD were *ANK3*, *ANKRD11*, *DPP6*, *FOXP1*, *GJA8*, *GRM8*, *IMMP2L*, *KCND2*, *MYT1L*, *NLGN1*, *NRXN1*, *SHANK2*, *SHANK3*, *SLC9A9*, *ST7*, *TBX1*, *TRAPPC9* and *TRIO*. The MD genes were *ANK3*, *ANKRD11*, *ATRX*, *CACNB2*, *CALHM3*, *CHMP2B*, *CHRM2*, *CSMD1*, *DEAF1*, *DLGAP3*, *DPP6*, *EBF3*, *FOXP1*, *GRID1*, *GRM8*, *IMMP2L*, *KAT6B*, *KIRREL3*, *LSAMP*, *MCTP1*, *MYT1L*, *NDP*, *NLGN1*, *NOS1AP*, *NRG1*, *NRXN1*, *RYR3*, *SHANK2*, *SHANK3*, *SKA2*, *SLC1A2*, *SLC1A3*, *SLC9A9*, *TENM4* and *TRAPPC9*. Other functional categories of genes are shown in Appendix A.

In pathway analysis, only the axon guidance pathway showed statistical significance when using KEGG or REACTOME functional databases. Appendix A illustrates the genes within the axon guidance pathway where DMCs were identified.

### 3.3. Comparison with ASD and Neurodevelopmental Gene Databases

In order to deeper understand the possible implications of genes with DMCs; these genes were compared with those in the databases related to ASD and neurodevelopment.

Sixty-four genes with DMCs have previously been associated with ASD and reported in the SFARI gene (Simons Foundation Autism Research Initiative) [28] database (Figure 3a). According to their classification score in four categories, four were syndromic, sixteen had a high confidence level, forty were strong candidates, four had suggestive evidence and one was found in two categories, syndromic and high confidence level, CELF2 (Figure 3b). The four syndromic ones and their associations with ASD are confirmed in several papers (*DMD*, *SLC1A2*, *TBX1* and *CELF2*) [24]. On the other hand, Leblond compared multiple data repositories with genes related to neurodevelopmental disorders (NDD), classifying them into different categories, including high-confidence-level genes related to NDD (HC-NDD) [29]. We also compared our gene set list with high-confidence-level NDD and SFARI genes, and 30 genes were common among three data sets: *IGF1*, *ARHGEF9*, *TRAPPC9*, *CNOT1*, *ZBTB20*, *SHANK3*, *NRXN1*, *DPP6*, *RERE*, *MYT1L*, *CELF2*, *CSNK1G1*, *SHANK2*, *DMD*, *KCND2*, *ATRX*, *TRIO*, *SETD2*, *CEP135*, *SLC1A2*, *ANKRD11*, *EBF3*, *TBX1*, *KCNQ2*, *BRSK2*, *DEAF1*, *FOXP1*, *CUX1*, *ANK3* and *KAT6B* (Figure 4). Four genes previously classified in the SFARI syndromic category (*DMD*, *SLC1A2*, *TBX1* and *CELF2*) and HC-NDD in Leblond’s dataset were maintained in this comparison, highlighting their relevance in ASD.

Otherwise, we found significant CpGs in 446 genes that have not been previously reported or associated with TEA, but these are still important (Appendix A).

## 4. Discussion

Recent evidence points to the role of altered DNA methylation in neurodevelopmental disorders, such as ASD. In this work, we analyzed differential DNA methylation between ASD patients and healthy controls in samples from buccal swabs. We found 853 DMCs in 509 protein-coding genes, which shows the value of buccal swab samples as a good non-invasive alternative to inaccessible brain tissue since buccal epithelial tissue derives from the same ectodermal layer as neurons do during neurodevelopment; thus, it has been suggested as a good alternative to reflect methylation profiles [19].

The top five genes with the most significant DMCs have not previously been linked to ASD (*ISM1*, *PTPRG*, *SLITRK4*, *CAP2* and *CYP26C1* genes). The gene *ISM1* (Isthmin 1) located on chromosome 20 has not been previously linked to psychiatric disorders or ASD. *ISM1* has been related to clefting and craniofacial patterning [30] and colorectal cancer [31,32] and probably plays a role in the negative regulation of angiogenesis [33]. The *PTPRG* gene product is part of the protein tyrosine phosphatase (PTP) family and has multiple functions in cell signaling such as cell growth, differentiation, oncogenic transformation and the mitotic cycle. In the central nervous system, PTPs play an important role in mediating differentiation and synaptic organization [34]. *CAP2* codifies for the cyclase-associated actin cytoskeleton regulatory protein 2, with a higher expression level in the brain. This gene was identified by its similarity to the gene for human adenylyl cyclase-associated protein. The function of the protein encoded by this gene is not completely understood; it is a spine-shaper protein located in the postsynaptic compartment involved in spine enlargement, and it appears to interact with adenylyl cyclase and actin [35,36]. *SLITRK4* located at Xq27.3 is a member of the SLIT and NTRK-like family that encode integral membrane proteins expressed mainly in the brain [36,37]. These proteins regulate neurite growth and synapse formation [38]. There is a report on Iranian patients with an altered expression of *SLITRK4* associated with fragile X syndrome caused by a repeat expansion in the 5′ untranslated region of the *FMR1* gene at the same chromosomal location [39]. The *CYP26C1* gene product is a member of the P450 enzyme cytochrome family involved in drug and lipid metabolism, including retinoic acid [40]. Although there is no evidence implicating this gene with ASD, retinoic acid signaling is crucial in neurodevelopment, and there are reports indicating the alteration of the retinoic acid pathway in schizophrenia.

In regard to enrichment analysis, autistic disorder was the third most enriched functional category, which could weigh the value of the alterations in methylation that we have found, in addition to the fact that another enriched category is mental disorders. Several authors have proposed that mental disorders, including ASD, share a common genetic background and common overlap processes and pathways based on their shared genetic variants and mutations [41,42,43,44,45]. Epigenetic alterations could additionally contribute to the proposed common genetic background of mental disorders.

Another ailment that could be linked to ASD is musculoskeletal diseases, since ASD patients show repetitive behaviors, hypermobility and other conditions. These patients frequently report fatigue, sensory alterations, coordination difficulties, osteopenia, dental diastasis, alterations in tooth eruption, flat feet and a pronated valgus [46]. A possible relationship between gene methylation and these affections, grouped as musculoskeletal diseases, must be studied thoroughly. Moreover, the fact that the axon guidance was the enriched pathway may be an indication of how important it is for neurodevelopment and the construction of neural circuits in the brain. Future studies should examine whether these epigenetic changes are a result of ASD-related abnormalities in neural connections.

On the other hand, by using hierarchical clustering with the 50 most significant DMCs, we found that ASD patient samples were almost completely separated from the controls, indicating a possible epigenetic profile in ASD patients. One case sample clustered within the controls. We reviewed the patient’s clinical characteristics as well as their methylation data. This patient does not show any distinctive clinical characteristics or any error in the DNA methylation data. So, the clustering reflects that it is not perfect, but it separated most patients from the controls, as reflected in other investigations [10]. Finding reproducible methylation patterns unique to the disorder could help in the early diagnosis of ASD patients; more research is needed with a higher number of samples to identify reliable and accurate methylation biomarkers for ASD. Furthermore, methylation biomarkers could provide insights into the underlying biology of ASD. The fact that we found the DMCs in genes described in the SFARI database supports the evidence that they are involved in the etiology of autism. SFARI is a specialized database of candidate genes involved in autism susceptibility by integrating genetic information from multiple research studies, which are selected by experts and constantly updated. The genes within this database are supported by evidence of their role in ASD; for example, gene variation in *SHANK2*, *SHANK3*, *FOXP1*, *ANKRD11*, *DEAF1*, *MYT1L* and *NRXN1* was reported in association with ASD in one of the largest exome sequencing studies [40]. More recently, it was associated with rare coding variants in *SHANK3*, *FOXP1*, *DEAF1*, *BRSK2*, *ZBTB20* and *EBF3* genes in ASD [47]. Moreover, the role of epigenetic regulation has been pointed out for *SHANK3*, as it alters DNA methylation in brain samples from ASD patients [48], which makes it a strong candidate as an epigenetic marker, with the advantage that its analysis is feasible using buccal swab samples. It is worth mentioning that most of the genes that we found with DMCs show similar methylation profiles between saliva or the oral epithelium and the brain, as reported in the IMAGE-CpG database [49]. On the other hand, DNA methylation alteration has been reported for *ANKRD11* in murine models for intrauterine exposure to a high-fat diet in relation to ASD, which could indicate a possible link between environmental and genetic factors. Future studies are needed to find out if genes that have been reported with altered expression in ASD, such as *FOXP1* or *SHANK2*, could be epigenetically regulated.

Other genes with DMCs, which are not included in SFARI, are candidates that participate in ASD susceptibility, and we need future studies to confirm their importance. Most of the DMCs were found within gene regions, and of these, around half were found in regulatory regions (UTRs and promoter regions), so we cannot rule out that some of these alterations in DNA methylation may have an effect on DNA methylation expression. Unfortunately, we do not have RNA to assess the expression in these samples. Additionally, the inclusion of some genes with DMCs in other databases of genes linked to ASD, such as the one reported by Leblond, reinforces the evidence that epigenetic processes are relevant for their regulation and their role as susceptibility genes for ASD.

## 5. Conclusions

In summary, the identification of altered DNA methylation in CpG sites from buccal swabs of ASD patients, notably within genes cataloged in the SFARI database, offers persuasive insights into the significance of these genes in the etiology of ASD. Epigenetic mechanisms are likely contributors to the regulation of these genes. Moreover, our findings endorse the feasibility of employing DNA methylation patterns as potential biomarkers and may contribute to the development of personalized approaches for the diagnosis and treatment of individuals with ASD.

## Figures and Tables

**Figure 1 brainsci-13-01420-f001:**
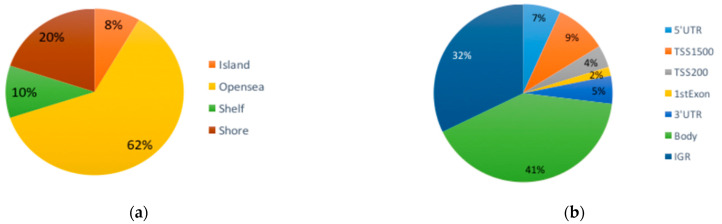
Differential DNA methylation in buccal swabs from children with autism. (**a**) Volcano plot showing CpG sites from the site-level test assessed via methylation differences and adjusted *p* values. All of the CpGs have FDR < 1 × 10^−10^; the CpGs in blue are differentially methylated and have more than 0.2 or less than −0.2. Manhattan plot showing the distribution of differentially methylated CpG sites across 22 autosomes and sexual chromosomes. The CpG sites marked in blue are the ones that reached a delta-beta ≥0.1 or ≤−0.1 and *p* value < 0.05. (**b**) Pie graph showing the percentage of different gene regions where CpGs were located. (**c**) Heat map from non-supervised analysis with case and control samples. Grouping of samples by methylation profiles separates most ASD patients from controls.

**Figure 2 brainsci-13-01420-f002:**
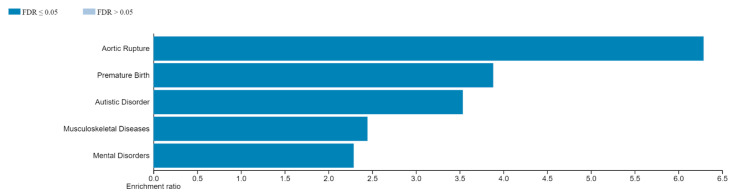
Functional categories obtained via gene enrichment analysis using WebGestalt. FDR ⋜ 0.05. These 5 categories were found with differentially methylated genes among cases and controls.

**Figure 3 brainsci-13-01420-f003:**
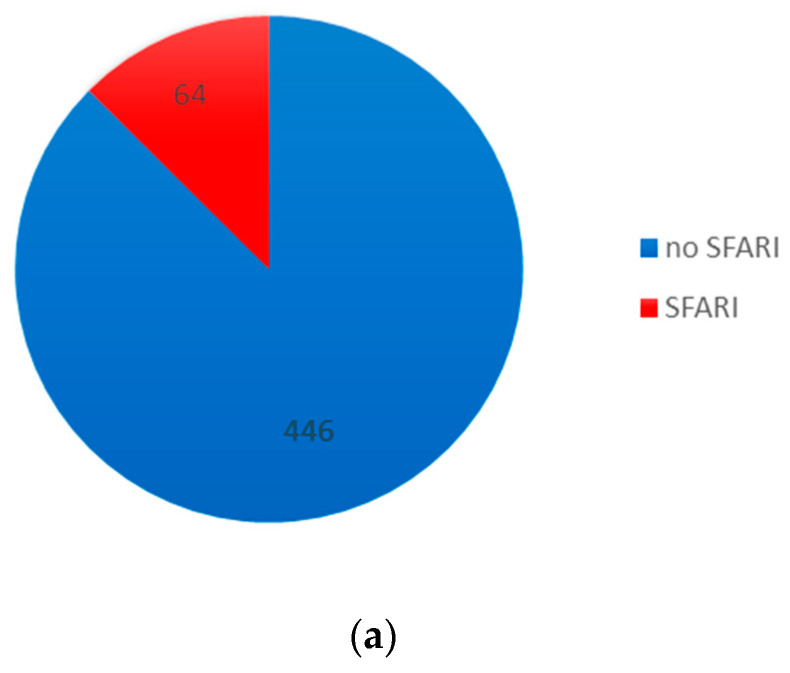
Total of genes identified in the methylation analysis. (**a**) DMCs were located in 510 genes, of which, 64 were in SFARI database and classified in four categories according to reported evidence. (**b**) SFARI gene categories were as follows: in red, 4 were syndromic; in orange, 16 had a high confidence level; in yellow, 40 were strong candidates; and in green, 4 had suggestive evidence.

**Figure 4 brainsci-13-01420-f004:**
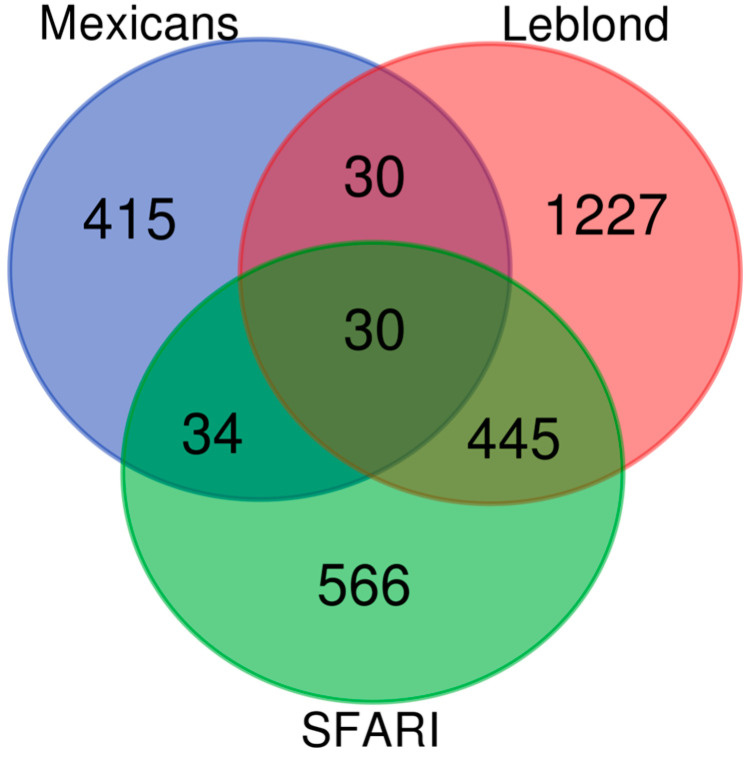
Venn diagram showing convergence of genes with 3 gene sets, Mexicans, SFARI and Leblond. Thirty genes were common among the 3 sets.

**Table 1 brainsci-13-01420-t001:** Description of the cases and control samples.

Samples	Mean Age (Years)	SD	Diagnosis	Therapy	Medication	Formal Schooling
Cases (29)	5.1	1.1	ASD	Yes:69% ND:31%	Yes:48.3% No:20.7% ND:31.0%	Yes:55.2% No:17.2% ND:27.6%
Controls (7)	5.7	1.1	None	None	None	Yes: 100%

Therapy: psychological, language, sensory and socialization; SD (standard deviation); ND (not described). Medication: risperidone, sertraline, methylphenidate, quetiapine, magnesium valproate, clonazepam, piracetam and aripiprazole. Scholarship: kindergarten, preschool and elementary school.

## Data Availability

All data are available under request following the personal privacy restrictions of participants. Credit to authors and institutions must be given.

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
