# Peer review of "Differential DNA Methylation from Autistic Children Enriches Evidence for Genes Associated with ASD and New Candidate Genes"

_brainsci, 2023, doi:10.3390/brainsci13101420_

Round 1
Reviewer 1 Report
Dear Author,
Thanks for submitting your research manuscript entitled “Differential DNA methylation from autistic children enriches evidence for genes associated with ASD and new candidate genes.".
Note: Before giving my final comments, as well as the final revision of this manuscript,
Firstly, the author needs to address the following comments scientifically.
Major concerns:-Please find out the following comments
Title, Abstract, and Introduction:
- Need complete revision for easy understand, and evaluate the manuscript in well defined manner.
- Old, outdated introduction; not relevant for publication.
- Rationale behind selection of DNA methylation from autistic children enriches evidence for genes associated with ASD is not clear.
- Incomplete and insufficient figures and lack of relevant preliminary and clinical data.
- The title needs to be very specific and is not acceptable in its current form.
- Lack of update as well old & outdated references with incomplete review design is another major concern.
- Outdated References quoted in text with important significance of DNA methylation from autistic children enriches evidence for genes associated with ASD is not clear throughout the manuscript.
- The rationale and purpose behind the correlation and selection of DNA methylation from autistic children enriches evidence for genes associated with ASD is not clear and incomplete throughout the manuscript.
- Abstract need improve.
- Authors must start their introduction direct with the correlation and rationale behind this study instead of writing about the common information regarding mechanism of DNA methylation from autistic children enriches evidence for genes associated with ASD.
- The abstract need improve. Irrational and fused with repetitions. Scientific output is not clear with this abstract.
- Irrational lines without scientific output is another major concern throughout the manuscript.
- The key messages and conclusion need improve. Author need to directly strike in scientific and readily manner.
- Provide the separate future perspective for this manuscript.
- The reviewer feels the author needs to elaborate and justify it with proper citations and strong evidence. The author fails to explain the relevant justification in the introduction as well as mentioned in the discussion part.
- In introduction, and other subparts of review, old and outdated references, 10 year and more than old references make it difficult to publish.
- A major drawback is a lack of clinical evidence, and the preliminary experimental data of DNA methylation from autistic children enriches evidence for genes associated with ASD.
- The reviewer found irrational and non-scientific justification in the abstract—introduction and the discussion part.
- Long paragraphs must be split into subheading according to content.
- Author must prepare one or more figures showing the involvement of rationale and theme focus on this research especially DNA methylation from autistic children enriches evidence for genes associated with ASD.
- Without any significant graphical abstract molecular pathways and figures make it difficult to further proceed.
Extensive editing of English language required
Author Response
Response to Reviewer 1 Comments
Dear reviewer,
Thank you very much for all the comments on our manuscript. Next, you will find the responses to your comments.
Best regards
Note: Before giving my final comments, as well as the final revision of this manuscript,
Firstly, the author needs to address the following comments scientifically.
Major concerns: Please find out the following comments
Title, Abstract, and Introduction:
Point 1: Need complete revision for easy understand, and evaluate the manuscript in well defined manner.
Response 1: We have modified the manuscript according to the observations and suggestions of all reviewers in order to clarify those points indicated.
Point 2: Old, outdated introduction; not relevant for publication.
Response 2: We have reviewed the introduction and verified that all definitions and data are up to date, and we have also modified a paragraph for better reading.
Point 3: Rationale behind selection of DNA methylation from autistic children enriches evidence for genes associated with ASD is not clear.
Response 3:
ASD is a multifaceted disorder, DNA methylation is a way to explore the complex interplay between genetics and epigenetics in the development of ASD. It offers insights into how specific genes may be regulated in individuals with ASD and provides a potential avenue for identifying biomarkers and therapeutic targets.
Point 4: Incomplete and insufficient figures and lack of relevant preliminary and clinical data.
Response 4: A table with clinical characteristics of samples was included in the new version.
Point 5: The title needs to be very specific and is not acceptable in its current form.
Response 5: We consider that the title reflects the results and conclusions of this paper. If the reviewers and the editor consider that we should change it, we will request the title change on the article submission platform.
Point 6: Lack of update as well old & outdated references with incomplete review design is another major concern.
Response 6: We have extensively reviewed the text and have updated the references for which there is more recent information. There are some basic references that are still relevant in the field or for which there is no change in the knowledge.
Point 7: Outdated References quoted in text with important significance of DNA methylation from autistic children enriches evidence for genes associated with ASD is not clear throughout the manuscript.
Response 7: We do not understand this observation, we have reviewed and updated the references that were outdated.
Point 8: The rationale and purpose behind the correlation and selection of DNA methylation from autistic children enriches evidence for genes associated with ASD is not clear and incomplete throughout the manuscript.
Response 8: We have made changes to the text to clarify the importance of our results. Changes in gene methylation may indicate that their expression may be regulated by epigenetic mechanisms, which provides new knowledge to understand the mechanism by which these genes are linked to ASD.
Point 9: Abstract need improve.
Authors must start their introduction direct with the correlation and rationale behind this study instead of writing about the common information regarding mechanism of DNA methylation from autistic children enriches evidence for genes associated with ASD.
Response 9: We have modified the abstract to address this issue.
Point 10: The abstract need improve. Irrational and fused with repetitions. Scientific output is not clear with this abstract.
Response 10: We have modified the abstract to address this issue.
Point 11: Irrational lines without scientific output is another major concern throughout the manuscript.
Response 11: We have exhaustively revised the text and have changed some phrases in the discussion to focus on the scientific knowledge on which they are based.
Point 12: The key messages and conclusion need improve. Author need to directly strike in scientific and readily manner.
Response 12: We have modified the conclusions to address this issue.
Point 13: Provide the separate future perspective for this manuscript.
Response 13: We have added a sentence in the discussion to indicate the future perspectives of this work
Point 14: The reviewer feels the author needs to elaborate and justify it with proper citations and strong evidence. The author fails to explain the relevant justification in the introduction as well as mentioned in the discussion part.
Response 14: We have modified the introduction and discussion sections to address this issue.
Point 15: In introduction, and other subparts of review, old and outdated references, 10 year and more than old references make it difficult to publish.
Response 15: We have extensively reviewed the text and have updated the references for which there is more recent information. There are some basic references that are still relevant in the field or for which there is no change in the knowledge.
Point 16: A major drawback is a lack of clinical evidence, and the preliminary experimental data of DNA methylation from autistic children enriches evidence for genes associated with ASD.
Response 16: We have modified the clinical description of the patients in the methods and results sections. We also added a table with a summary of the clinical characteristics of the patients
Point 17: The reviewer found irrational and non-scientific justification in the abstract—introduction and the discussion part.
Response 17: We have modified the abstract and introduction sections to address this issue
Point 18: Long paragraphs must be split into subheading according to content.
Response 18: We have revised the text and it has been sent to a professional English editing service.
Point 19: Author must prepare one or more figures showing the involvement of rationale and theme focus on this research especially DNA methylation from autistic children enriches evidence for genes associated with ASD.
Response 19: Figure 3 shows the genes that have previously been related to ASD and that were shown to have alterations in methylation in the present work, which contributes to increasing the evidence of the participation of these genes in ASD. We have modified the discussion section to clarify this information.
Point 20: Without any significant graphical abstract molecular pathways and figures make it difficult to further proceed.
Response 20: We have added a file with the graphical abstract as requested by the reviewer and editor.
Reviewer 2 Report
The manuscript describes the study of methylation patterns of genes potentially associated with ASD. This study made it possible not only to confirm the contribution of methylation of known genes, but also to identify a number of new ones.
I have no comments on the manuscript - the study has been carried out at a good methodological level, the data obtained are new and original.
There are only some comments that may allow the authors to improve their work:
1. Are there any data on the significance of DNA methylation in certain regions (TSS, 5'UTR, Body, 3'UTR) of a gene on its transcriptional activity? Is it possible to analyze the identified genes from this point of view? Perhaps, it is worth adding reasoning about this to the Discussion.
2. Are there any correlations between the significance of individual genes and the number of DNA methylation sites within them? For some, the number of methylation sites is 3 or more (CUX1, TRIO, NXN, etc.). Perhaps, it is worth adding reasoning about this to the Discussion.
3. it is better to bring all references to the same format (Wong, 2014) - line 64.
Author Response
Response to Reviewer 2 Comments
Dear reviewer,
Thank you very much for all the comments on our manuscript. Next, you will find the responses to your comments.
Best regards
Point 1: 1. Are there any data on the significance of DNA methylation in certain regions (TSS, 5'UTR, Body, 3'UTR) of a gene on its transcriptional activity? Is it possible to analyze the identified genes from this point of view? Perhaps, it is worth adding reasoning about this to the Discussion.
Response 1:
We agree with your comment, we have modified the results section and added a discussion paragraph to address the issue of the distribution of the DMCs in the different gene regions.
Point 2: 1. Are there any correlations between the significance of individual genes and the number of DNA methylation sites within them? For some, the number of methylation sites is 3 or more (CUX1, TRIO, NXN, etc.). Perhaps, it is worth adding reasoning about this to the Discussion.
Response 2:
There was no correlation between the number of DMCs in a gene and the significance. We have added sentences in the methods, results and discussion sections to point out that when several DMCs were present in the same gene, they did not clustered to reach significance to be considered a differentially methylated region (DMR).
Point 3: it is better to bring all references to the same format (Wong, 2014) - line 64.
Response 3: The reference was corrected in the manuscript.
Reviewer 3 Report
I suggest the following revision:
The text has some typos (extra spaces). Review carefully. Line 68, 143, 175, 176.
Line 86: The study does not seem to be population-based, but rather a convenience sample. I suggest that it be described how the 29 cases were selected from the consortium samples. Also, how the controls were selected and if there was any kind of matching.
I also suggest describing whether the patients with autism also had dysmorphias and whether they performed any other type of genetic ingestion.
The paper is succinct and describes what is necessary to support the idea proposed by the authors.
Author Response
Response to Reviewer 3 Comments
Dear reviewer,
Thank you very much for all the comments on our manuscript. Next, you will find the responses to your comments.
Best regards
Point 1: The text has some typos (extra spaces). Review carefully. Line 68, 143, 175, 176.
Response 1: The manuscript was carefully reviewed and those errors were corrected.
Point 2: Line 86: The study does not seem to be population-based, but rather a convenience sample. I suggest that it be described how the 29 cases were selected from the consortium samples. Also, how the controls were selected and if there was any kind of matching.
Response 2: The paragraph was corrected in the first section of materials and methods.
Point 3: I also suggest describing whether the patients with autism also had dysmorphias and whether they performed any other type of genetic ingestion.
Response 3: We included a table with the main description of patients. Patients do not have genetic dysmorphia or other genetic syndromes.
Reviewer 4 Report
The manuscript of Morales-Marín et al. presents the results of a methylation microarray performed with buccal swabs of ASD children and controls. I believe the authors have rich results and material, but there are several aspects that need to be improved in the manuscript prior to its publication. My main suggestions are:
1. Sample characteristics should be presented as a Table. Clinical data regarding ASD children should be provided, especially because one of the patients clustered with the controls.
2. Both case and control groups (inclusion and exclusion criteria) should be better detailed. Data presented in Lines 115-117 should be moved to the Methods section.
3. What is the SRS interview? These details must be provided because not all the readers are familiar with the questionnaires applied.
4. Why did the authors use such a small sample size in the control group when compared to the cases?
5. Are all the 509 genes of the protein-coding type?
6. Enrichment analyses are not described in the Methods section.
7. The authors have great material in their hands, hence I believe they could have provided more informative figures and tables than the pizza plots. Especially regarding the results presented in sections 3.2 and 3.3.
8. The authors mentioned one patient was clustered with the control group. Why do the authors believe the methylation profile of this patient in particular was different from the others? This should be discussed in the Discussion section.
9. The authors mention using the GSEA method in the Discussion. When I read the manuscript, I thought they had performed an overrepresentation analysis in WebGestalt. If they used GSEA, the parameters used must be better explained.
10. I suggest the authors focus on their results rather than discussing the use of buccal swabs. It is well-established that blood and buccal samples provide great transcriptomic and epigenetic information, especially considering the difficulty of obtaining brain samples.
11. Minor point: In Line 50 it is written ADS instead of ASD.
12. I suggest the authors consider submitting their raw methylation data to public repositories, such as Gene Expression Omnibus (GEO). It encourages scientific reproducibility and transparency.
Lines 51-52: The sentence appears to be incomplete.
Line 68: Did the authors mean "ASD animal models induced by valproic acid"?
Lines 72-75: The sentence is too long. Did the authors mean the search for methylation markers should not be restricted to brain samples?
Line 78: Did the authors mean "by their application"?
Line 122: I suggest to rephrase to: "DMC location corresponded to 509 genes".
Line 129-130: I suggest rephrasing to "Most of the genes had a single DMC, with only 33 genes presenting two or more DMCs."
Line 131: Correct to "heatmap".
Line 166-167: I did not understand if "they" refers to the genes or DMCs.
Line 168: Correct to "Sixty-four".
Author Response
Response to Reviewer 4 Comments
Dear reviewer,
Thank you very much for all the comments on our manuscript. Next, you will find the responses to your comments.
Best regards
Point 1: Sample characteristics should be presented as a Table. Clinical data regarding ASD children should be provided, especially because one of the patients clustered with the controls.
Response 1:
Thank you for your comment, we added a new table with the sample characteristics.
Point 2: Both case and control groups (inclusion and exclusion criteria) should be better detailed. Data presented in Lines 115-117 should be moved to the Methods section.
Response 2: We have changed the materials seccion to include a better detailed description of patients and controls criteria for inclusion and exclusion.
Lines 115-117 were moved to the methods section as you requested.
Point 3: What is the SRS interview? These details must be provided because not all the readers are familiar with the questionnaires applied.
Response 3: The description of SRS (Social Responsiveness Scale) was added in the manuscript.
Point 4: Why did the authors use such a small sample size in the control group when compared to the cases?
Response 4: The sample size is one of the limitations of our study, we have added a sentence in the discussion section to point this out.
Point 5: Are all the 509 genes of the protein-coding type?
Response 5: The description was complemented in the first paragraph in results section as follows:
“DMC were located in 509 protein coding genes, 278 intergenic regions and 15 non-coding RNA genes, distributed throughout the whole genome, as seen in the Manhattan plot (Figure 1a lower panel and Table S1).”
Point 6: Enrichment analyses are not described in the Methods section.
Response 6: The next paragraph was added to the methods section.
For enrichment analysis we used WebGestalt, a functional enrichment analysis web tool per se. Its main function is to translate gene lists into biological insights. Parameters chosen were: organism of interest: Homo sapiens; method of interest: ORA (over-representaion analysis); functional database: disease. Parameters for the enrichment analysis: FDR method: Benjamini-Hochberg; Significance level: FDR < 0.05.
This paragraph was added in the section “Functional analysis of DMCs”.
Point 7: The authors have great material in their hands, hence I believe they could have provided more informative figures and tables than the pizza plots. Especially regarding the results presented in sections 3.2 and 3.3.
Response 7: We have added a figure to better explain enriched pathway analysis.
Point 8: The authors mentioned one patient was clustered with the control group. Why do the authors believe the methylation profile of this patient in particular was different from the others? This should be discussed in the Discussion section.
Response 8: We have reviewed the patient's clinical characteristics, as well as methylation data. The patient does not show any distinctive clinical characteristic nor any error in the DNA methylation data, so the clustering reflects that it is not perfect but it separates most patients from controls.
Point 9: The authors mention using the GSEA method in the Discussion. When I read the manuscript, I thought they had performed an overrepresentation analysis in WebGestalt. If they used GSEA, the parameters used must be better explained.
Response 9: We used the ORA method for enrichment analysis, it was corrected in the discussion section. The ORA parameters were added in the Methods section.
Point 10: I suggest the authors focus on their results rather than discussing the use of buccal swabs. It is well-established that blood and buccal samples provide great transcriptomic and epigenetic information, especially considering the difficulty of obtaining brain samples.
Response 10: We have modified the discussion section to address this point.
Point 11: Minor point: In Line 50 it is written ADS instead of ASD.
Response 11: It was corrected in the manuscript.
Point 12: I suggest the authors consider submitting their raw methylation data to public repositories, such as Gene Expression Omnibus (GEO). It encourages scientific reproducibility and transparency.
Response 1: If necessary we submit the raw data to a public repository, meanwhile data are available under request as we indicated in the manuscript.
Point 13: Lines 51-52: The sentence appears to be incomplete.
Response 13: The sentence was removed from the text.
Point 14: Line 68: Did the authors mean "ASD animal models induced by valproic acid"?
Response 14: That is the sentence meaning, it was corrected in the manuscript.
Point 15: Lines 72-75: The sentence is too long. Did the authors mean the search for methylation markers should not be restricted to brain samples?
Response 15: The sentence was corrected as follows:
Although it has been pointed out that methylation patterns are tissue-specific, the search for methylation markers was not restricted only to brain samples.
Postmortem brain samples from children are particulary extremely difficult to obtain and are just aviable in brain bionbanks.
Point 16: Line 78: Did the authors mean "by their application"?
Response 16: Yes, that’s the right meaning. It was corrected in the manuscript.
Point 17: Line 122: I suggest to rephrase to: "DMC location corresponded to 509 genes".
Response 17: It was corrected as suggested.
Point 18: Line 129-130: I suggest rephrasing to "Most of the genes had a single DMC, with only 33 genes presenting two or more DMCs."
Response 18: The phrase was changed in the main text.
Point 19: Line 131: Correct to "heatmap".
Response 19: It was corrected in the text.
Point 20: Line 166-167: I did not understand if "they" refers to the genes or DMCs.
Response 20: It refers to “genes”. The correction was made in the manuscript.
Point 21: Line 168: Correct to "Sixty-four".
Response 21: It was corrected in the manuscript.
Round 2
Reviewer 1 Report
Dear author,
after careful revision, manuscript revised successfully
Reviewer 4 Report
The authors performed all the alterations requested and the manuscript was greatly improved.